# New Vortex-Synchronized Matrix Solid-Phase Dispersion Method for Simultaneous Determination of Four Anthraquinones in Cassiae Semen

**DOI:** 10.3390/molecules24071312

**Published:** 2019-04-03

**Authors:** Luyi Jiang, Jie Wang, Huan Zhang, Caijing Liu, Yiping Tang, Chu Chu

**Affiliations:** 1College of Pharmaceutical Science, Zhejiang University of Technology, Hangzhou 310014, China; 15700082652@163.com (L.J.); saynoproblem@163.com (H.Z.); 15958027383@163.com (C.L.); 2College of Materials Science and Engineering, Zhejiang University of Technology, Hangzhou 310014, China; wangjie930724@163.com

**Keywords:** vortex-synchronized matrix solid-phase dispersion, crab shells, ionic liquids, anthraquinones, Cassiae Semen

## Abstract

In this study, a green ionic-liquid based vortex-synchronized matrix solid-phase dispersion (VS-MSPD) combined with high performance liquid chromatography (HPLC) method was developed as a quantitative determination method for four anthraquinones in Cassiae Semen. Two conventional adsorbents, C_18_ and silica gel were investigated. The strategy included two steps: Extraction and determination. Wasted crab shells were used as an alternative adsorbent and ionic liquid was used as an alternative solvent in the first step. Factors affecting extraction efficiency were optimized: A sample/adsorbent ratio of 2:1, a grinding time of 3 min, a vortex time of 3 min, and ionic liquid ([Domim]HSO_4_, 250 mM) was used as eluent in the VS-MSPD procedure. As a result, the established method provided satisfactory linearity (R > 0.999), good accuracy and high reproducibility (RSD < 4.60%), and it exhibited the advantages of smaller sample amounts, shorter extraction time, less volume of elution solvent, and was much more environmental-friendly when compared with other conventional methods.

## 1. Introduction

As seafood consumption has increased, shellfish cultivation has emerged as an expanding economic activity in the world, accounting for more than 40% of all marine aquaculture production [1]. Crab, is one of the top ten highest consumed sea food products and continues to compete well against other seafood proteins. However, only a small portion (20–30%) of the weight of crab has been consumed as food on tables, the rest is generally considered to be garbage and is discarded at will [2]. Consequently, a public health problem caused by the wastage of crab shell resources and the deterioration of the environment, which is due to their low biodegradation has arisen [3]. There is an urgent need to develop methods for treating wasted crab shells and making this bio waste valuable.

Crab shell powder (CSP) of marine waste was used for biofungicide production about ten years ago [4,5]. Recently, a good way of recycling wasted crab shells is its application in the removal of pollutants as an adsorbent [1,2,6,7,8,9,10,11,12,13]. It has been reported that crab shells consist of chitin (or chitosan), calcium carbonate along with some proteins. Among them, chitosan, possessing available amino groups on its polymeric chain, shows high metal binding capability and has been used for the adsorption of heavy metals (Cu, Pb, Cd, Ni, Cr, Au, Se, V, Eu, Co, Ce) [6,7,8,9,10,11,12], As(V) [14], and other substances such as phosphate [13] and dye (Congo Red) [1,2]. Moreover, the advantages of being used as an adsorbent include the low cost, great mechanical strength, rigid structure, ability to withstand drastic conditions and high biocompatibility [14,15]. So far, however, there have only been a few reports on the extraction of bioactive compounds in a complex matrix.

Cassiae Semen, derived from the ripe seed of *Cassia obtusifolia* L. or *C. tora* L. is an officially edible and medicinal plant in China [16]. It is commonly drunk as a kind of healthy tea for its function in liver protection, bacteriostasis, catharsis and immune adjustment, eyesight improvement, diuresis, antitumor, and antioxidation in many southeast Asian countries [17,18,19]. It has also been used as a slimming tea to aid metabolic processes by helping detoxify the body [20]. A variety of bioactive anthraquinones, including aurantio-obtusin, chryso-obtusin, obtusin, aloe-emodin, rhein, emodin, chrysophanol, physcion and glucosides, have been reported in Cassiae Semen [19,21]. Among them, chrysophanol and aurantio-obtusin were selected as markers for quality control of Cassiae Semen in the Chinese Pharmacopoeia (2015 ed). However, chrysophanol is non-specific in Cassiae Semen, therefore more bioactive markers are in urgent need to be used for its quality evaluation. The development of efficient and sensitive methodologies to determine those bioactive compounds is also needed.

Several extraction techniques, including heating reflux extraction or ultrasonic extraction, have been reported in the literature for the extraction of anthraquinones from Cassiae Semen [17,18,19,20,22,23,24,25]. Those methods either need a high volume of toxic solvents, such as methanol or chloroform, or a long extraction time. Additionally, the analytes are exposed to the chemicals in a consecutive clean-up step. Recently, using environmentally-friendly solvents, keeping acceptable accuracy and extraction efficiency, has become the trend of the development of modern methods. Therefore, VS-MSPD technique based on a matrix solid phase dispersion by using vortex agitation instead of the solid phase extraction step, seems to be a good choice. Several studies have shown that it leads to acceptable results with the advantage of high speed, simplicity, and a low cost material in complex matrices [26,27,28,29,30,31,32].

MSPD was developed for the treatment of solid, semisolid and highly viscous samples by dispersing the matrix in a solid phase adsorbent [33]. Recent advances in MSPD include the fact that using cheaper, greener solid supports, nanoparticles or environmental-friendly eluents instead of traditional ones [34,35,36,37]. In this study, a green vortex-synchronized matrix solid-phase dispersion (VS-MSPD) combined with a HPLC method has been developed to determine four anthraquinones, including aurantio-obtusin, chryso-obtusin, obtusifolin and emodin in Cassiae Semen samples. The experimental procedure consisted of two steps, which were extraction and elution. In the extraction step, solid waste from the sea food industry, CSP, was firstly applied as an alternative adsorbent to extract natural components in the food matrix. Moreover, green ionic liquids (ILs)such as [Bmim]PF_6_, [Domim]Cl, [Domim]NO_3_ and [Domim]HSO_4_ were used instead of a harmful organic solvent for extraction for the first time. The main objective of this study was to develop a method that needed smaller sample amounts, a shorter extraction time, a smaller volume of elution solvent, and was much more environmental-friendly. The developed method has shown great potential on rapid extraction and determination of natural products from complex samples.

## 2. Results and Discussion

### 2.1. Characterization of Crab Shell Powder (CSP)

SEM analysis was carried out to observe the morphology of CSP, as show in Figure 1. The size of CSP was less than 80 μm, which agreed with the size of a standard sieve with 200 mesh. Additionally, from the high magnification shown in Figure 1b, the morphology of CSP was very loose and porous indicating the high surface area, which benefits the adsorption process.

A FT-IR spectrum was also applied to evaluate CSP, as shown in Figure 2. There were six main peaks in the spectra. The sharp peaks located at 1414, 874, 713 cm^−1^ corresponded to CO_3_^2−^, which confirmed the existence of calcite [38]. The strong peak around 3344 cm^−1^ was O-H stretching vibration [39] and N-H stretching vibration of chitin [40]. The peak around 2961 cm^−1^ was ascribed to C-H stretching vibration [41]. The peak around 1070 cm^−1^ was due to C-O stretching vibration of carbohydrate [42]. This implied that chitin was the main composition of CSP.

### 2.2. VS-MSPD Optimization

MSPD was developed to extract compounds from solid, semisolid and highly viscous samples by disrupting and dispersing the matrix in a solid phase adsorbent [33]. The sample components were dissolved by the bonded organic phase and better dispersed on the surface of the support. The surface area of the extracted sample was increased, and the sample was dispersed in the organic phase according to their respective polarities. Furthermore, to obtain the optimal extraction efficiency of the four anthraquinones on VS-MSPD, some factors were investigated, including the type of adsorbent, sample/adsorbent ratio, the grinding time, the vortex time, the concentration and the type of ionic liquid and volume of ionic liquid. All of these parameters were evaluated in detail and each test was repeated in triplicate. Analysis of variance (ANOVA) and Duncan’s test were conducted among different groups by SPSS software. The comparisons with *P* < 0.05 were considered significant.

The extraction efficiency of an MSPD procedure depends on the type of adsorbent used [36]. The mechanism of adsorption, the choice of eluent and the interfering compounds remaining on its surface are affected by the different physicochemical properties of adsorbent. In the current experiment, three different adsorbents, including CSP and two conventional adsorbents, C_18_ and silica gel, were investigated. The mixture with the sample/adsorbent ratio of 1:1 was ground for 3 min and eluted with 250 mM [Domim]NO_3_ aqueous solution by 3 min-vortex. As shown in Figure 3a, C_18_ provided a relatively poor response for all of the anthraquinones, which may be ascribed to the hydrophobic characteristics and the non-polar nature of the reversed-phase C_18_ particles. Silica gel with abundant Si-O-Si and Si-OH groups accelerated the rate of forming hydrogen bonds between sorbent affording higher extraction efficiency than C_18_ to four anthraquinones [37]. CSP showed similar extraction efficiency to silica gel. It might be explained that after base treatment, chitosan was the main residue contained in CSP, which possessed available -NH_2_ and -OH groups on its polymeric chain and therefore showed high binding capability with anthraquinones [2]. Moreover, the cellulose-like backbone of CSP may also help to adsorb anthraquinones [13]. For the purpose of making the bio waste valuable, CSP was selected as an optimal adsorbent in the following study.

The sample/adsorbent ratio is a significant factor affecting extraction efficiency since the contact surface/active sites on the adsorbent surface should be sufficient to trap the target compounds [43]. The mixture of CSP and sample was ground for 3 min and eluted with 250 mM [Domim]NO_3_ aqueous solution by 3 min-vortex. It was seen from Figure 3b that the ratio 2:1 (mg of samples/mg of adsorbent) was found to be optimal for all investigated anthraquinones. Moreover, the extraction efficiency decreased when the ratio changed to 1:1, 1:2 or 1:3, which might be ascribed to the stronger adsorption capacity and the harder elution with the increasing amount of CSP. The decreased extraction efficiency might also be due to increasing hydrogen bonding and electrostatic interaction between the four target compounds and CSP [36]. Thus, the sample/adsorbent ratio of 2:1 was finally chosen in the subsequent experiment.

An entire adsorption requires a sufficient contraction time to get a homogeneous mixing [37]. The mixture of CSP and sample (2:1) was ground at different times and vortexed with a 250 mM [Domim]NO_3_ aqueous solution over 3 min. Figure 3c vividly depicts a different extraction efficiency with an increased grinding time. The result indicated that the extraction efficiency enhanced when the grinding time increased from 1 min to 3 min, which may be ascribed to the much stronger contraction between four analytes and CSP. However, the response of all the target compounds decreased while the grinding time prolonged to 4 min, which might have been due to a long grinding time leading to an overly strong extraction of the four anthraquinones, which increased the difficulty of elution. Therefore, grinding for 3 min was used for further investigations.

An adequate vortex time can elute four analytes from CSP completely [37]. The mixture of CSP and sample with the ratio of 2:1 was ground for 3 min and vortexed with a 250 mM [Domim]NO_3_ aqueous solution for 3 min. As shown in Figure 3d, the extraction efficiency raised when the vortex time increased from 1 min to 3 min. However, the extraction efficiency of a 4 min-vortex was no more significantly increased compared to that of a 3 min-vortex, which indicated that 3 min of vortex was enough in this method. Therefore, a vortex time of 3 min was selected in the following study.

The use of solvents for both the environment and human beings associated with sample preparation is one of the goals of green Analytical Chemistry. Ionic liquids possess unique chemico-physical properties, and are a kind of green solvent which has been applied widely in chemical synthesis, catalysis, separation sciences and electrochemistry [36]. The mixture of CSP and sample (2:1) was ground for 3 min and eluted with 250 mM different kinds of ILs aqueous solution by 3 min-vortex.In this work, four types of ILs, including [Bmim]PF_6_, [Domim]Cl, [Domim]HSO_4_ and [Domim]NO_3_, were studied to obtain a satisfactory extraction efficiency. As seen in Figure 4, [Bmim]PF_6_ had the best elution efficiency of aurantio-obtusin, but the response of chryso-obtusin, obtusifolin and emodin was not very good, which might have been due to the diverse polarity between [Bmim]PF_6_ and the three target compounds. The other three ILs, [Domim]Cl, [Domim]HSO_4_ and [Domim]NO_3_, although consisting of the same long-chain cation, had different elution efficiencies on four target compounds, in which the extraction efficiency of [Domim]HSO_4_ and [Domim]NO_3_ was better than [Domim]Cl. That might be ascribed to the weaker electrostatic interaction between the anion Cl^−^ and analytes. The results showed that a better extraction efficiency could be obtained with [Domim]HSO_4_ as the elution solvent, which might have accounted for the dominating π–π and hydrogen bonding interactions from the HSO_4_^−^ anion and the stronger electrostatic interaction between the anion HSO_4_^−^ and analytes [37]. Furthermore, interactions with analytes might also be possible due to the ion-dipole and the inclusion complexation, which might affect the extraction efficiency [36]. Finally, [Domim]HSO_4_ was selected as the eluent in further experiments.

The concentration of IL was another crucial factor in the elution procedure, because a suitable concentration can result in a satisfactory extraction of the analytes. The mixture of CSP and sample according to the ratio of 2:1, was ground for 3 min and eluted with different concentrations of [Domim]HSO_4_ aqueous solution by 3 min-vortex. It was found in Figure 5 that the peak areas of four target compounds increased from 150 mM to 250 mM, which might have been due to the solubility and elution capacity of the solvent, which were enhanced with an increased ionic liquid concentration. Upon further increases of [Domim]HSO_4_ concentration, the extraction efficiency was not much more significantly increased, which might be accountable to the increase in the solution viscosity resulting in the difficulty with the mass transfer of the anthraquinones from the Cassiae Semen matrix into [Domim]HSO_4_ aqueous solution [36]. On the basis of the above results, the concentration of 250 mM was chosen as the best compromise.

### 2.3. Method Validation

Method validation was characterized on the basis of linearity, LODs, LOQs, precision, reproducibility and recovery. The results are demonstrated in Table 1 and Table 2. The calibration curves of four analytes were built in a form of *y* = a*x* + b with peak area (*y*) and the concentration of the analyte (*x*). Good correlation coefficients (R > 0.999) were obtained for all analytes. The LOD values were estimated to be 1.40, 2.00, 1.04 and 0.44 μg·mL^−1^ for aurantio-obtusin, chryso-obtusin, obtusifolin and emodin, respectively. The LOQs were based on S/N = 10, which ranged from 2.20 to 13.30 μg·mL^−1^. Additionally, the precision values of intra-day and inter-day shown in Table 1 were less than 3.59% and 2.90%, respectively. Moreover, the reproducibility values of the method were estimated to be 1.79%, 4.60%, 2.41% and 3.03% for aurantio-obtusin, chryso-obtusin, obtusifolin and emodin, respectively. The stability values were in the 2.00–3.15% range, which indicated the sample solution was stable in 24 h. As listed in Table 2, the average recoveries of four analytes were all in the range of 91.30–106.40% with relative standard deviations (RSDs) of 0.98–3.98%. These results indicated that crab shell-based VS-MSPD using ILs as eluent coupled with HPLC method exhibited satisfactory linearity, good accuracy and high reproducibility.

### 2.4. Application to Real Samples

The developed VS-MSPD method was successfully applied for the analysis of four anthraquinones in Cassiae Semen. As shown in Table 3, the contents of aurantio-obtusin, chryso-obtusin, obtusifolin and emodin in Cassiae Semen were in the range of 0.40–2.95 mg·g^−1^, 0.1–0.33 mg·g^−1^, 0.12–0.48 mg·g^−1^ and 0.02–0.07 mg·g^−1^, respectively. The contents of the four anthraquinones were reported as mean ± standard deviation from five independent batches. Analysis of variance (ANOVA) and Duncan’s test were conducted among different batches by SPSS software. With the value of *P* < 0.05, the difference between different batches was considered significant. A typical HPLC chromatogram demonstrating the separation of four analytes is depicted in Figure 6. It was interesting to find the contents of four target compounds in the processed samples were higher than those in the raw materials, which was consistent with the results reported by Guo [24]. These results demonstrated that the VS-MSPD method was feasible for the analysis of complicated semen food samples.

### 2.5. Comparison with Other Methods

A comparison of the extraction method, analytes, sample amounts, solvents, and extraction time with other reported methods was performed to estimate the analytical performance with the developed method. The results are summarized in Table 4. It was easily found that the new developed VS-MSPD method required smaller sample amounts, a shorter extraction time and less volume of elution solvent. More importantly, the use of recycled wasted crab shells as an adsorbent and the ionic liquid as elution solvent were more environmentally-friendly than other methods. Overall, on the basis of the aforementioned results, this green VS-MSPD method combined with HPLC is rapid, sensitive, simple and promising for the extraction and determination of anthraquinones in complex food matrices.

## 3. Materials and Methods

### 3.1. Chemicals and Reagents

The ionic liquids used in the study included 1-butyl-3-methylimidazolium hexafluorophosphate ([Bmim]PF_6_), 1-dodecyl-3-methyl chloride imidazole ([Domim]Cl), 1-dodecyl-3-methyl-1-*H*-imidazolium hydrogensulfate ([Domim]HSO_4_), 1-dodecyl-3-methyl-1-*H*-imidazolium nitrate ([Domim]NO_3_), were offered by Chengjie Chemical Co., Ltd. (Shanghai, China). C_18_ and silica gel were purchased from Shanghai Chengya Chemical Co., Ltd. (Shanghai, China). Standard substances including aurantio-obtusin, chryso-obtusin, obtusifolin and emodin were supplied from Chengdu Must Bio-Technology Co., Ltd. (Chengdu, China). Their purities were all higher than 98% and their structures are shown in Figure 7. Purified water was purchased from Wahaha Group Ltd. (Shanghai, China). HPLC grade methanol and acetonitrile were obtained from American Tedia Co., Ltd. (Shanghai, China). Formic acid was of analytical reagent grade from Shanghai Lingfeng Chemical Reagent Co., Ltd. (Shanghai, China). All reagents for HPLC were filtrated through a 0.22 μm filter.

### 3.2. Preparation and Characterization of CSP

Crab shells were obtained from a local restaurant supplier (Hangzhou, China), and then rinsed with tap water in order to remove other debris and slime. After that, they were washed again with purified water and dried in an electric oven at 60 °C for about 12 h to a constant weight. The dried crab shells were crushed into powder and sieved with 100 meshes. Then, base treatment was conducted using 20% (wt) aqueous NaOH at 100 °C for about 1h in order to remove redundant protein, and the powders were washed with purified water and dried in an electric oven at 60 °C for about 12 h. The treated powders were crushed again, passed over 200 meshes, and stored in dry vacuum packs for ready use as an adsorbent. A scanning electron microscope (TESCAN VEGA 3 SBH) was used to examine the surface morphology of CSP. FT-IR spectra was investigated to analyze CSP by a Nicolet 6700 FT-IR Spectrometer (TA Instruments, Wilmington, DE, USA) in the range of 4000 to 400 cm^−1^.

### 3.3. HPLC Analysis

Chromatographic analyses of aurantio-obtusin, chrys-oobtusin, obtusifolin and emodin were performed with Agilent 1260 series (Agilent, Santa Clara, CA, USA) using an ultraviolet (UV) detection set at 440 nm. The mobile phase consisted of 95% acetonitrile-5% water (*v*/*v*) (A) and 5% acetonitrile-95% water (*v*/*v*) (B), which all contained 0.1% formic acid with a flow rate of 1 mL·min^−1^. The gradient elution was set as follows: 0–15 min, 5–55% A; 15–18 min, 55–55% A; 18–26 min, 55–80% A; 26–27 min, 80–100% A; 27–31 min, 100–100% A. A Shimadzu Inersustain C_18_ column (4.6 × 250 mm, 5 μm) was used with the column temperature hold set at 25 °C. The injection volume was 20 μL.

### 3.4. Preparation of Standard Solution

The mixed stock solution was prepared in methanol respectively including 123.6 μg·mL^−1^ of aurantio-obtusin, 40.0 μg·mL^−1^ of chryso-obtusin, 41.6 μg·mL^−1^ of obtusifolin and 13.2 μg·mL^−1^ of emodin. The prepared standard solution was stored in a freezer at 4 °C.

### 3.5. VS-MSPD Procedure and Normal MSPD Procedure

Dried Cassiae Semen material was crushed and passed through 50 meshes. About 20 mg of sample powder and 20 mg of adsorbent (CSP, C_18_ and silica) were slightly placed into an agate mortar and ground with a pestle for 3 min. The obtained mixtures were transferred into a 10 mL centrifuge polypropylene tube. Then different types of ionic liquids were added, each were 1 mL. The mixture was entirely whirled with vortex for 3 min and filtrated through 0.22 μm filter. The filter eluent was collected in a 1.5 mL centrifuge tube. After centrifuging at 13,000 rpm for 6 min, the supernatant was collected for the HPLC analysis.

Approximately 20 mg of Cassiae Semen powder and 20 mg of CSP were placed into an agate mortar and blended using a pestle until a visually homogeneous mixture was obtained (for 3 min). Following complete dispersal, the mixture was carefully transferred into a 1 mL SPE column with a frit at the bottom. Then a second frit was pressed slightly on the top of the blend with a syringe plunger. The target analytes were eluted afterwards with 1 mL of [Domim]Cl aqueous solution (250 mM) in a vacuum. The eluent was collected in a 1.5 mL centrifuge tube and centrifuged at 13,000 rpm for 6 min. The supernatant was subjected to the HPLC analysis.

### 3.6. Heating Reflux Extraction (HRE)

HRE was performed based on the method documented in Chinese Pharmacopoeia (2015 ed) with minor modification [22]. Briefly, 0.2 g of Cassiae Semen powder was dissolved in 10 mL of methanol. Then the mixture was extracted by heating reflux for 2 h. After the mixture was extracted, the weight loss of the solution was compensated with methanol. The final solution was passed through a 0.22 μm membrane filter, and analyzed by HPLC.

### 3.7. Validation of Analytical Procedure

Linearity was evaluated by analyzing a series of concentrations of mixed standard solution (including aurantio-obtusin, chryso-obtusin, obtusifolin and emodin). After analysis, four calibration curves were constructed by plotting peak areas (*y*) versus concentration (*x*, mg·mL^−1^). Moreover, correlation coefficients were obtained by least squares regression. The limits of detection (LODs) and limits of quantification (LOQs) of the proposed method were assessed by the analysis of the four standards at a signal-to-noise ratio of 3 and 10, respectively. Intra-day precision RSDs were determined by six replicate injections of the mixed standard solution analyzed on the same day. Inter-day RSDs were calculated with the detection of the mixed standard solution for three consecutive days. Repeatability of the method was examined with the determination of six replicates of the same sample. Stability was obtained by evaluating a sample solution at 2, 4, 8, 16 and 24 h. Recovery was determined by standard addition method. A certain number of standards including low and high concentration were added into a certain amount of Cassiae Semen powder respectively. The samples were extracted and analyzed with the proposed method.

## 4. Conclusions

In this study, a green sorbent, recycling wasted crab shell, was applied as the dispersing agent and environmentally-friendly extraction solvent IL ([Domim]HSO_4_) was used as an eluent of MSPD for the simultaneous extraction of four anthraquinones in Cassiae Semen samples. Compared to the Chinese National Standard (Chinese Pharmacopiea 2015 ed) and other conventional methods, the presented method exhibited advantages of green, smaller sample amounts, a shorter extraction time and less volume of elution solvent. Furthermore, satisfactory linearity, good accuracy and high reproducibility were achieved, indicating that the developed VS-MSPD method which is a quantitative determination method of the anthraquinones can provide an efficient and powerful tool for the extraction of natural products in real samples.Thus, the method is significant in the quality control of edible and medicinal Cassiae Semen.

## Figures and Tables

**Figure 1 molecules-24-01312-f001:**
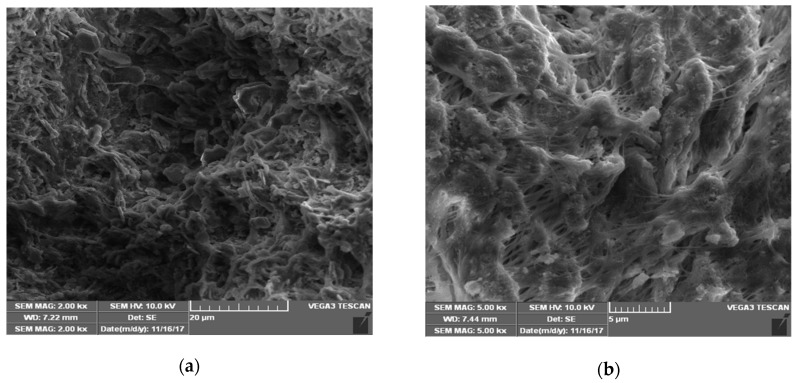
SEM images of CSP at (**a**) low and (**b**) high magnification.

**Figure 2 molecules-24-01312-f002:**
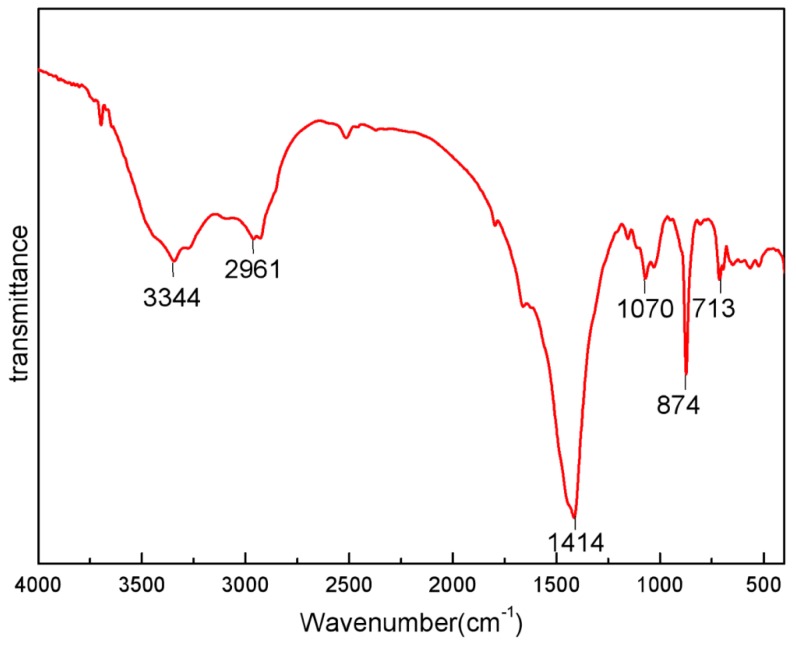
FT-IR spectrum for CSP.

**Figure 3 molecules-24-01312-f003:**
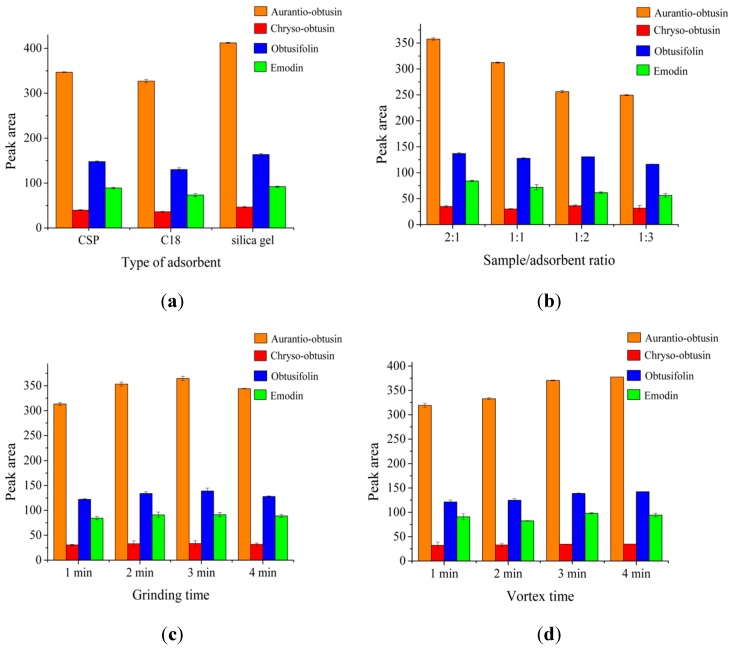
(**a**)Effect of different adsorbents in VS-MSPD process; (**b**) effect of sample/adsorbent ratio in VS-MSPD process; (**c**) effect of grinding time in VS-MSPD process; (**d**) effect of vortex time in the VS-MSPD process.

**Figure 4 molecules-24-01312-f004:**
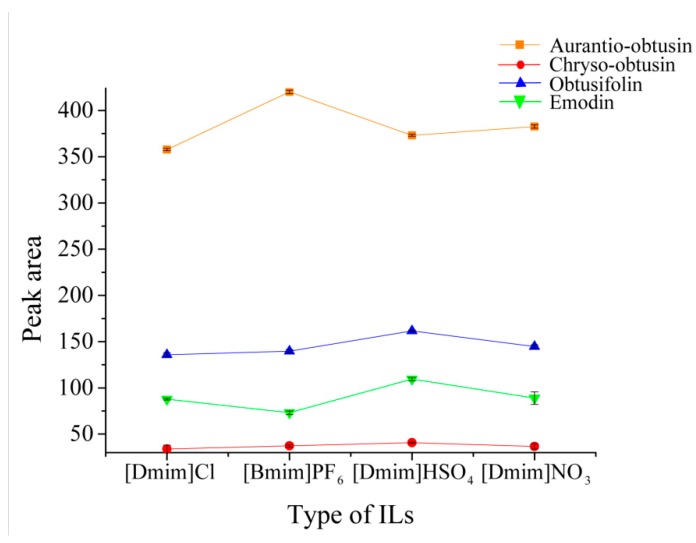
Effect of different ILs in VS-MSPD process.

**Figure 5 molecules-24-01312-f005:**
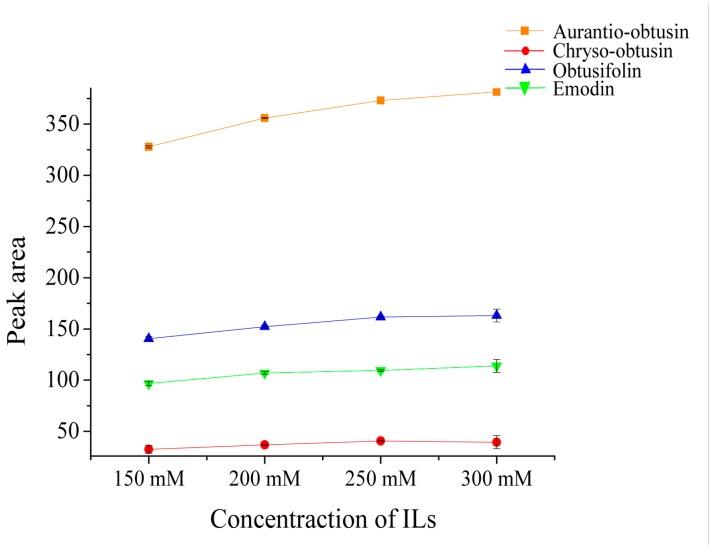
Effect of concentration ILs in VS-MSPD process.

**Figure 6 molecules-24-01312-f006:**
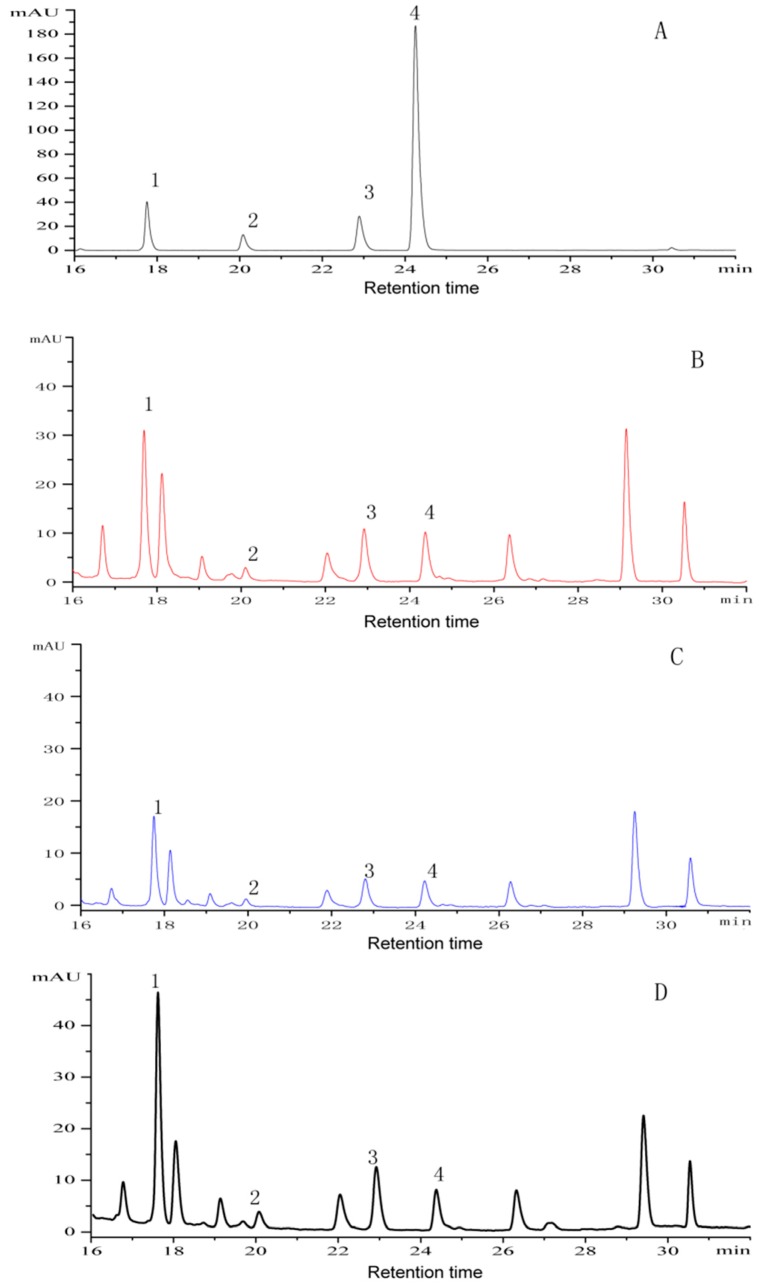
HPLC chromatograms of four anthraquinones in Cassiae Semen (**A**) standard solution of four anthraquinones, (**B**) extracted using heating reflux extraction, (**C**) extracted using normal miniaturized MSPD, (**D**) extracted using VS-MSPD. Peaks: 1 = Aurantio-obtusin, 2 = Chryso-obtusin, 3 = Obtusifolin, 4 = Emodin.

**Figure 7 molecules-24-01312-f007:**
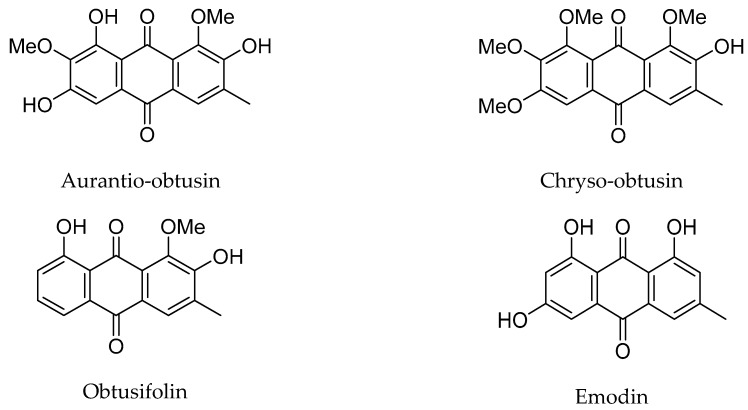
Structures of the four standard substances.

**Table 1 molecules-24-01312-t001:** Linear regression data, limit of detection (LOD), limit of quantitation (LOQ), precision, reproducibility and stability of the entire analytical method.

Analytes	Linear Regression Data		Range (μg mL^−1^)	LOD (μg mL^−1^)	LOQ (μg mL^−1^)	Precision (RSD%)		Reproducibility (RSD%)	Stability (RSD%)
	Calibration Curve	R				Intra-Day (*n* = 6)	Inter-Day (*n* = 3)		
Aurantio-obtusin	y = 10.048x + 6.8409	0.9997	3.09–123.60	1.40	4.00	1.26	2.50	1.79	2.47
Chryso-obtusin	y = 2.5094x − 1.4986	0.9997	2.00–40.00	2.00	13.30	3.59	2.90	4.60	2.75
Obtusifolin	y = 13.724x + 0.7768	0.9999	1.04–41.60	1.04	3.50	1.17	2.90	2.41	3.15
Emodin	y = 35.053x + 1.3954	0.9998	1.32–13.20	0.44	2.20	1.91	2.90	3.03	2.00

**Table 2 molecules-24-01312-t002:** The recoveries of four analytes.

Analytes	Original (µg)	Spiked (µg)	Found (µg)	Recoveries (%)	RSD (%)
Aurantio-obtusin	18.22	9.11	27.33	94.34	3.78
		18.22	36.44	103.68	2.85
		27.33	45.55	99.67	2.67
Chryso-obtusin	8.40	4.20	12.60	92.90	3.97
		8.40	16.80	100.00	3.98
		12.60	21.00	98.22	3.88
Obtusifolin	5.86	2.93	8.79	91.30	2.54
		5.86	11.72	102.72	1.48
		8.79	14.65	106.40	2.93
Emodin	1.54	0.77	2.31	94.60	3.72
		1.54	3.08	105.19	0.98
		2.31	3.85	99.60	3.39

**Table 3 molecules-24-01312-t003:** Contents (mg·g^−1^) (mean ± SD) of four compounds in five batches of Cassiae Semen samples by VS-MAPD method.

Sample Number	Aurantio-Obtusin	Chryso-Obtusin	Obtusifolin	Emodin
1 ^a^	1.48 ± 0.18	0.11 ± 0.01	0.23 ± 0.03	0.03 ± 0.01
2 ^b^	2.93 ± 0.03	0.30 ± 0.04	0.45 ± 0.03	0.06 ± 0.01
3 ^a^	0.42 ± 0.02	0.21 ± 0.02	0.12 ± 0.01	0.02 ± 0.00
4 ^b^	2.92 ± 0.02	0.23 ± 0.04	0.47 ± 0.02	0.06 ± 0.00
5 ^a^	1.69 ± 0.20	0.11 ± 0.02	0.27 ± 0.04	0.02 ± 0.01

^a^ Raw Cassiae Semen. ^b^ Fried Cassiae Semen.

**Table 4 molecules-24-01312-t004:** Comparison of VS-MSPD method with other methods in the determination of compounds in Cassiae Semen.

No.	Extraction Method	Analytes	Sample Amounts (g)	Type of Solvent	Solvent Volume (mL)	Extraction Time (min)	Reference
1	Heating reflux extraction	chrysophanol; aurantio-obtusin	0.5	methanol	50	120	[22]
2	Heating reflux extraction	aurantio-obtusin; obtusin; chrysophanol; emodin; chrysophanol; physcion	0.5	90% methanol	50	120	[23]
3	Ultrasonic extraction	emodin; rhein; autrantio-obtusin	0.2	chloroform	60	40	[19]
4	Heating reflux extraction	aurantio-obtusin; obtusifolin; questin; SC-1; rhein; emodin; SC-2	50	75% methanol	500	60	[24]
5	Accelerated solvent extraction (ASE)	aurantio-obtusin; aloe-emodin; rhein; emodin; physcion; chrysophanol	0.2	acetonitrile	11	8	[20]
6	Ultrasonic extraction	emodin; aloe-emodin; rhein	2.0	ethanol-chloroform (1:1)	75	90	[25]
7	Microwave assisted extraction	aloe-emodin; rhein; emodin; chrysophanol; physcion	0.1	10% Genapol X-080	6	3	[17]
8	Heating reflux extraction	2-gluco-aurantioobtusin; casside; rhein; torachrysoneglucosides; physcion; 2-gluco-chrysoobtusin; aurantio-obtusin; chryso-obtusin; 1-desmethylobtusin; obtusin; aloe-emodin; emodin; chrysophanol	0.5	70% ethanol	50	180	[18]
9	VS-MSPD	aurantio-obtusin; chryso-obtusin; obtusifolin; emodin	0.02	250 mM [Domim]HSO_4_	1	10	This work

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
