# Peer review of "New Vortex-Synchronized Matrix Solid-Phase Dispersion Method for Simultaneous Determination of Four Anthraquinones in Cassiae Semen"

_molecules, 2019, doi:10.3390/molecules24071312_

Round 1

Reviewer 1 Report

The authors reported the utilisation of seafood byproduct, which is a good topic for sustainable development. Some results showed some potential applications. However, the authors need to improve the manuscript significantly, especially to provide some evidence to support some statements and provide statistical analysis for the results.

Line 54: add ‘has’ before ‘also’.

Line 66: environmentally-friendly;

Line 70: add ‘high’ before ‘speed’;

Line 72-76: There were no supporting evidence to support this statement. Additionally, nanoparticle based MSPD is a recent development in the field but the authors did not mention any of this point. It is suggested to add some related documents to support the statement, for instance, Food Chemistry, 275, 390-396; Food Control, 88, 9-14; Food Control, 74, 112-120; Food Chemistry, 217, 303-310.

Line 85: change ‘is’ to ‘was’. The following lines should be changed as well. The experimental results should be reported in past tense.

Figures 3-6: some Figures should be combined together. Furthermore, significant analysis among different groups should be conducted and labelled.

Table 3: significant analysis among different batches should be conducted and labelled.

Discussion section should be improved. The current version mentioned results but did not discuss them fully.  

Author Response

From

College of Pharmaceutical Sciences, Zhejiang University of Technology

Hangzhou, 310014, China

Tel.: +86-571-88320613

Fax: +86-571-88320913

E-mail: chuchu@zjut.edu.cn

To

Reviewer 1

Dear Reviewer 1,

Thank you for your valuable and helpful comments. The manuscript has been revised in accordance with the reviewer’s comments. Enclosed please find the revised manuscript entitled "Recycling wasted crab shells as a potential adsorbent in vortex-synchronized matrix solid-phase dispersion method for simultaneously determination of four anthraquinones in Cassiae Semen" here attached. 

Hopefully, the response will satisfy the reviewer’s criticism. If you have any further question regarding the revised version, please do not hesitate to let me know.

Thank you for the kind advices.

Yours sincerely,

Chu Chu

College of Pharmaceutical Sciences, Zhejiang University of Technology

Response to reviewer #1.

1) Line 54: add has before also.

Reply: Thanks very much for your kind suggestion. It has been revised as seen in Line 52.

2) Line 66: environmentally-friendly.

Reply: We appreciate your kind advice. It has been changed, as seen in Line 64.

3) Line 70: add high before speed.

Reply: We really thank the kind suggestion of the reviewer. It has been revised and high has been added in Line 68.

4) Line 72-76: There were no supporting evidence to support this statement. Additionally, nanoparticle based MSPD is a recent development in the field but the authors did not mention any of this point. It is suggested to add some related documents to support the statement, for instance, Food Chemistry, 275, 390-396; Food Control, 88, 9-14; Food Control, 74, 112-120; Food Chemistry, 217, 303-310.

Reply: We appreciate your useful comment. Some related references, especially nanoparticle based MSPD methods (references 34-35), have been added in Line 72. See references 34-37.

Reference 34. He, Z., Yang, H. Colourimetric detection of swine-specific DNA for halal authentication using gold nanoparticles. Food Control 2018, 88, 9-14. DOI: 10.1016/j.foodcont.2018.01.001.

Reference 35. Yu, X., Yang, H. Pyrethroid residue determination in organic and conventional vegetables using liquid-solid extraction coupled with magnetic solid phase extraction based on polystyrene-coated magnetic nanoparticles. Food Chem. 2017, 217, 303-310.DOI: 10.1016/j.foodchem.2016.08.115.

Reference 36. Xu, J.J.; Yang, R.; Ye, L.H.; Cao, J.; Cao, W.; Hu, S.S.; Peng, L.Q. Application of ionic liquids for elution of bioactive flavonoid glycosides from lime fruit by miniaturized matrix solid-phase dispersion. Food Chem. 2016, 204, 167-175. DOI: 10.1016/j.foodchem.2016.02.012.

Reference 37. 37.Du, K.Z.; Li, J.; Bai, Y.; An, M.R.; Gao, X.M.; Chang, Y.X. A green ionic liquid-based vortex-forced MSPD method for the simultaneous determination of 5-HMF and iridoid glycosides from Fructus Corni by ultra-high performance liquid chromatography. Food Chem. 2018, 244, 190-196.

DOI: 10.1016/j.foodchem.2017.10.057.

5) Line 85: change isto was. The following lines should be changed as well. The experimental results should be reported in past tense.

Reply: We really thank the kind observation of the reviewer. The experimental results have been reported in past tense.

6) Figures 3-6: some Figures should be combined together. Furthermore, significant analysis among different groups should be conducted and labelled.

Reply: Thanks very much for your kind suggestion. Figures 3-6 have been combined and significant analysis among different groups has been conducted. The P values have been calculated and labeled in discussion section, as seen Lines 107-109.

7) Table 3: significant analysis among different batches should be conducted and labeled.

Reply: Thanks very much for your comment. Significant analysis among different batches has been conducted. The P value among five batches has been calculated and labeled in discussion section. See Lines 213-216. 

8) Discussion section should be improved. The current version mentioned results but did not discuss them fully.

Reply: We really thank the kind comment of the reviewer. Refer to published papers (References 36-37), discussion section has been fully revised. The revised part has been noted by red font. 

Reviewer 2 Report

Manuscript entitled “Recycling wasted crab shells as a potential adsorbent in vortex-synchronized matrix solid-phase dispersion method for simultaneously determination of four anthraquinones in Cassiae Semen” submitted by Luyi Jiang, Jie Wang, Huan Zhang, Caijing Liu, Yiping Tang, Chu Chu, can be accepted for publishing in the Molecules Journal, after major revisions.

In this study, a green ionic-liquid based vortex-synchronized matrix solid-phase dispersion (VS-MSPD) combined with HPLC method has been developed to determine those four anthraquinones. The manuscript presents original results that are consistent, but their organization and interpretation must be significantly improved.

            Here is a list of my specific comments:

Title: The title should be changed      because it is not in agreement with the manuscript content. A possible      alternative could be: New vortex-synchronized      matrix solid-phase dispersion method for simultaneously determination of      four anthraquinones in Cassiae Semen.

Page 1, Abstract: This section needs to be rewritten to      make it clearer. Include here the following mentions: (i) this is a      quantitative determination method for these four anthraquinones, (ii) the      experimental procedure include two steps; (iii) the wasted crab shells is      used as alternative adsorbent in the first step.

Page 2, line 72: “Recent advance in matrix solid-phase      dispersion…”. This method should be more detailed described.

Page 2,      line 76: “Solid waste from sea food industry, CSP, was firstly applied…”.      Replace this paragraph with “The experimental procedure consists of two      steps, first…” (or similar). The same formulation should be included also      in Abstract to be clearer. Also, at the end of Introduction, the main      objectives of this study should be reformulated.

Page 2,      2. Results and Discussion: (i) Add here a paragraph to explain the      principle of matrix solid-phase dispersion method. (ii) Explain the      selection of wasted crab shells for such experimental studies.

Page 3,      line 89: Replace “And FT-IR spectra were applied…” with “FT-IR spectra      were also used …” (or similar).

Page 3,      2.2. VS-MSPD optimization: In all this section, the exact experimental      conditions should be mentioned (ionic liquid, extraction time, etc.).

Page 4,      lines 100-101: “including CSP and two conventional adsorbents, C18 and      silica gel, were investigated.” This aspect should be also mentioned in      Abstract.

All other      sections from the manuscript should be discussed from the perspective of      quantitative determination method of the anthraquinones.

Author Response

From

College of Pharmaceutical Sciences, Zhejiang University of Technology

Hangzhou, 310014, China

Tel.: +86-571-88320613

Fax: +86-571-88320913

E-mail: chuchu@zjut.edu.cn

To

Reviewer 2

Dear Reviewer 2,

Thank you for your valuable and helpful comments. The manuscript has been revised in accordance with the reviewer’s comments. Enclosed please find the revised manuscript entitled "Recycling wasted crab shells as a potential adsorbent in vortex-synchronized matrix solid-phase dispersion method for simultaneously determination of four anthraquinones in Cassiae Semen" here attached. We are grateful for the valuable and helpful comments of the reviewers.

Hopefully, the response will satisfy the reviewer’s criticism. If you have any further question regarding the revised version, please do not hesitate to let me know.

Thank you for the kind advices.

Yours sincerely,

Chu Chu

College of Pharmaceutical Sciences, Zhejiang University of Technology

Response to reviewer #2.

1) Title: The title should be changed because it is not in agreement with the manuscript content. A possible alternative could be: New vortex-synchronized matrix solid-phase dispersion method for simultaneously determination of four anthraquinones in Cassiae Semen.

Reply: We agree with your comment. The title has been changed as the reviewer suggested, as seen in Lines 2-4.

2) Page 1, Abstract: This section needs to be rewritten to make it clearer. Include here the following mentions: (i) this is a quantitative determination method for these four anthraquinones, (ii) the experimental procedure include two steps; (iii) the wasted crab shells is used as alternative adsorbent in the first step.

Reply: Thanks very much for your advice. The abstract section has been rewritten, taking into account the suggestion of reviewer.

3) Page 2, line 72: “Recent advance in matrix solid-phase dispersion…”. This method should be more detailed described.

Reply: The reviewer is right. MSPD method has been described more detailed. Please see Lines 69-72.

4) Page 2, line 76: “Solid waste from sea food industry, CSP, was firstly applied…”.Replace this paragraph with The experimental procedure consists of two steps, first…” (or similar). The same formulation should be included also in Abstract to be clearer. Also, at the end of Introduction, the main objectives of this study should be reformulated.

Reply: Thanks very much for your comment. The sentence has been rewritten as asked by the reviewer, as seen in Lines 74-75. Moreover, the Abstract and Introduction also have been changed.

5) Page 2, 2. Results and Discussion: (i) Add here a paragraph to explain the      principle of matrix solid-phase dispersion method. (ii) Explain the selection of wasted crab shells for such experimental studies.

Reply: It has been revised and a paragraph has been added as suggested by the reviewer, as seen in Lines 99-103. The reason of selection of wasted crab shell for such experimental has been explained in Lines 123-125.

6) Page 3,line 89: Replace “And FT-IR spectra were applied…” with FT-IR spectra were also used …” (or similar).

Reply: Thanks very much for your suggestion. The sentence has been changed. See Line 91.

7) Page 3,2.2. VS-MSPD optimization: In all this section, the exact experimental      conditions should be mentioned (ionic liquid, extraction time, etc.).

Reply: We appreciate your kind comment. In all VS-MSPD optimization sections, the exact experimental conditions were added. See Lines 114-115, 130-132, 140-141,148-150, 157-159 and 175-177.

8) Page 4, lines 100-101: “including CSP and two conventional adsorbents, C18 and      silica gel, were investigated.This aspect should be also mentioned in      Abstract.

Reply: Thanks very much for your suggestion. The aspect has been added in Lines 16-17 and 112-114.

9)All other sections from the manuscript should be discussed from the perspective of quantitative determination method of the anthraquinones.

Reply: We really thank your comment. The perspective of quantitative determination method of the anthraquinones has been added in abstract and conclusion sections, as seen in Lines 14-17 and 320-324.

Round 2

Reviewer 1 Report

The manuscript has been improved.

Reviewer 2 Report

All my previous remarks and comments have been considered in this new version of the manuscript. In my opinion, the revised manuscript meets the criteria and can be published as original paper in Molecules Journal.